# Learning with Differentiable Perturbed Optimizers

**Quentin Berthet**
Google Research,
Brain Team - Paris, France
qberthet@google.com

**Mathieu Blondel**
Google Research,
Brain Team - Paris, France
mblondel@google.com

**Olivier Teboul**
Google Research,
Brain Team - Paris, France
oliviert@google.com

**Marco Cuturi**
Google Research,
Brain Team - Paris, France
cuturi@google.com

**Jean-Philippe Vert**
Google Research,
Brain Team - Paris, France
jpvert@google.com

**Francis Bach**
INRIA - DI, ENS,
PSL Research University Paris
francis.bach@inria.fr

## Abstract

Machine learning pipelines often rely on optimization procedures to make discrete decisions (e.g., sorting, picking closest neighbors, or shortest paths). Although these discrete decisions are easily computed, they break the back-propagation of computational graphs. In order to expand the scope of learning problems that can be solved in an end-to-end fashion, we propose a systematic method to transform optimizers into operations that are differentiable and never locally constant. Our approach relies on stochastically perturbed optimizers, and can be used readily together with existing solvers. Their derivatives can be evaluated efficiently, and smoothness tuned via the chosen noise amplitude. We also show how this framework can be connected to a family of losses developed in structured prediction, and give theoretical guarantees for their use in learning tasks. We demonstrate experimentally the performance of our approach on various tasks.

## 1 Introduction

Many applications of machine learning benefit from the possibility to train by gradient descent compositional models using end-to-end differentiability. Yet, there remain fields where discrete decisions are required at intermediate steps of a data processing pipeline (e.g., in robotics, graphics or biology). This is the result of many factors: discrete decisions provide a much sought-for interpretability, and discrete solvers are built upon decades of advances in combinatorial algorithms [47] for quick decisions (e.g., sorting, picking closest neighbors, exploring options with beam-search, or with shortest paths problems). These discrete decisions can easily be computed in a forward pass. Their derivatives with respect to inputs are however degenerate: small changes in the inputs either yield no change or discontinuous changes in the outputs. Discrete solvers thus break the back-propagation of computational graphs, and cannot be incorporated in end-to-end learning.

In order to expand the set of operations that can be incorporated in differentiable models, we propose and investigate a new, systematic method to transform discrete optimizers into differentiable operations. Our approach builds upon the method of stochastic perturbations, the theory of which was developed and applied to several tasks of machine learning recently; see [27]. In a nutshell, we perturb the inputs of a discrete solver with random noise, and consider the perturbed solutions of the problem. The method is both easy to analyze theoretically and simple to implement. We show that the formal expectations of these perturbed solutions are never locally constant and everywhere differentiable, with successive derivatives being expectations of simple expressions.

**Related work.** Our work is part of growing efforts to modify operations to make them differentiable. Several works have studied the introduction of regularization in the optimization problem to make the argmax differentiable. These works are usually problem-specific, since a new optimization problem needs to be solved. Examples include assignments [3], optimal transport [11, 17], differentiable dynamic programming [36], differentiable submodular optimization [19], ranking [18, 10], and top-$k$ [56, 5, 31]. A generic approach is SparseMAP [38], based on Frank-Wolfe or active-set algorithms for solving, and on implicit differentiation for Jacobian computation. Like our proposal, SparseMAP only requires access to a linear maximization oracle. However, it is sequential in nature, while our approach is trivial to parallelize. In [42], a related approach is proposed, with squared euclidean regularization on the probability space (rather than the mean space). To solve some computational difficulties on projecting on such a large space, top-$k$ oracles are leveraged. In [4], implicit differentiation on solutions of convex optimization is analyzed. They express the derivatives of the argmax exactly, leading to zero Jacobian almost everywhere when optimizing over polytopes. A possible perspective on efforts to handle discrete operators is to consider on one hand changes to the loss function, such as the max-margin loss [51, 20] used in recent architectures [46]; and on the other hand gradient estimation, through a proxy of the operator that often relies on regularization or perturbation. Our proposition uses stochastic perturbations. As part of the latter, Vlastelica et al. [54] propose to interpolate in a piecewise-linear manner between locally constant regions. The aim is to keep the same value for the Jacobian of the argmax for a large region of inputs, allowing for zero Jacobians as well. In subsequent work, this has been coupled with improvement on loss functions in specific applications [44, 45]. Our work provides contribution to both these approaches.

An example of expectation of a perturbed argmax, commonly known as the "Gumbel trick", dates back to Gumbel [23], and random choice models [33, 35, 22]. It is exploited in online learning and bandits to promote exploration, and induce robustness to adversaries (see, e.g., [2] for a survey). It is used for action spaces that are combinatorial in nature [37], used together with a softmax to obtain differentiable sampling [29, 34], to learn parsing trees in a differentiable fashion [15, 16] and with distributions from extreme value theory [7].

The use of perturbation techniques as an alternative to MCMC techniques for sampling was pioneered by Papandreou & Yuille [40]. They are used to compute expected statistics arising in gradients of conditional random fields. They show exactness for the fully perturbed (but intractable case) and propose "low-rank" perturbations as an approximation. These results are extended in [25], proving that the expected maximum with low-rank perturbations provides an upper-bound on the log partition, and replacing the log partition in conditional random fields loss by that expectation. Their results, however, are limited to discrete product spaces. New lower bounds on the partition function are derived in [26], as well as a new unbiased sequential sampler for the Gibbs distribution based on low-rank perturbations. These results were further refined in [21] and [39], and these bounds further studied in [49], who proposed a doubly stochastic scheme. Apart from [32], who use a finite difference method, we are not aware of any prior work using perturbation techniques to differentiate through an **argmax**. As reviewed above, all papers focus on (approximately) sampling from the Gibbs distribution, upper-bounding the log partition function, or differentiating through the **max**.

**Contributions.** We make the following contributions:

- We propose a new general method transforming discrete optimizers, inspired by the stochastic perturbation literature. This versatile method applies to any blackbox solver without ad-hoc modifications.

- Our stochastic smoothing allows **argmax** differentiation, through the formal perturbed maximizer. Its Jacobian is well-defined and non-zero everywhere, thereby avoiding vanishing gradients.

-The successive derivatives of the perturbed maximum and argmax are expressed as simple expectations, which are easy to approximate with Monte-Carlo methods.

- Our method yields natural connections to the recently-proposed Fenchel-Young losses by Blondel et al. [9]. We show that the equivalence via duality with regularized optimization makes these losses natural. We propose a doubly stochastic scheme for their minimization in learning tasks, and we demonstrate our method on structured prediction tasks, in particular ranking (permutation prediction), for which conditional random fields and the Gibbs distribution are intractable.

## 2 Perturbed maximizers

Given a finite set of distinct points $\mathcal{Y} \subset \mathbb{R}^d$ and $\mathcal{C}$ its convex hull, we consider a general discrete optimization problem parameterized by an input $\theta \in \mathbb{R}^d$ as follows:

$$F(\theta) = \max_{y \in \mathcal{C}} \langle y, \theta \rangle, \; y^*(\theta) = \arg\max_{y \in \mathcal{C}} \langle y, \theta \rangle.$$
(1)

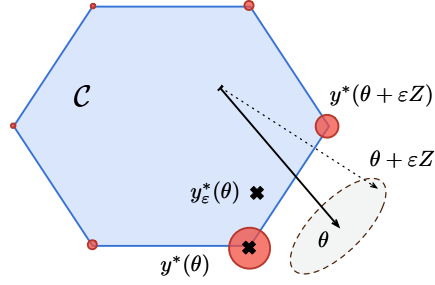

Figure 1: Stochastic smoothing yields a perturbed optimizer $y_\varepsilon^*$ in expectation.

As we discuss below, this formulation encompasses a variety of discrete operations commonly used in machine learning. In all cases, $\mathcal{C}$ is a convex polytope and these problems are linear programs (LP). For almost every $\theta$, the argmax is unique, and $y^*(\theta) = \nabla_\theta F(\theta)$. While widespread, these functions do not have the convenient properties of blocks in end-to-end learning architectures, such as smoothness or differentiability. In particular, $\theta \mapsto y^*(\theta)$ is piecewise constant: its gradient is zero almost everywhere, and undefined otherwise. To address these issues, we simply add to $\theta$ a random noise vector $\varepsilon Z$, where $\varepsilon > 0$ is a temperature parameter and $Z$ has a positive and differentiable density $\mathrm{d}\mu(z) \propto \exp(-\nu(z))\mathrm{d}z$ on $\mathbb{R}^d$, so that $y^*(\theta + \varepsilon Z)$ is almost surely (a.s.) uniquely defined. This induces a probability distribution $p_\theta$ for $Y \in \mathcal{Y}$ given by $p_\theta(y) = P(y^*(\theta + \varepsilon Z) = y)$; see Figure 1.

This creates a general and natural model on the variable $Y$, when observations are solutions of optimization problems, with uncertain costs. It enables the modeling of phenomena where agents chose an optimal $y \in \mathcal{C}$ based on uncertain knowledge of $\theta$. We view this as a generalization, or alternative to the Gibbs distribution, rather than an approximation thereof.

Taking expectations with respect to the random perturbation leads to smoothed versions of $F$ and $y^*$:

**Definition 2.1.** For all $\theta \in \mathbb{R}^d$, and $\varepsilon > 0$, we define the **perturbed maximum** as

$$F_\varepsilon(\theta) = \mathbf{E}[F(\theta + \varepsilon Z)] = \mathbf{E}[\max_{y \in \mathcal{C}} \langle y, \theta + \varepsilon Z \rangle],$$

and, the **perturbed maximizer** as

$$y_\varepsilon^*(\theta) = \mathbf{E}_{p_\theta(y)}[Y] = \mathbf{E}[\arg\max_{y \in \mathcal{C}} \langle y, \theta + \varepsilon Z \rangle] = \mathbf{E}[\nabla_\theta \max_{y \in \mathcal{C}} \langle y, \theta + \varepsilon Z \rangle] = \nabla_\theta F_\varepsilon(\theta).$$

Models of random optimizers for linear problems with perturbed inputs are the subject of a wide litterature in machine learning, under the name of "perturb-and-MAP" [40, 25], and perturbed leader method in online learning [24, 30, 1]. We refer to it here as the *perturbed model*.

**Broad applicability.** Many operations used in machine learning can be written in the form of Eq. (1) and are thus part of our framework. Indeed, for any score function $s : \mathcal{Y} \to \mathbb{R}$, the problem $\max_{y \in \mathcal{Y}} s(y)$, can at least be written as a linear program (LP) in Eq. (1), for some embedding of the set $\mathcal{Y}$. We emphasize that the LP structure need not be known to use the perturbed maximizers. In our experiments, we focus on the following three tasks (see Appendix B for more examples).

*Maximum.* The max function from $\mathbb{R}^d$ to $\mathbb{R}$, that returns the largest among the $d$ entries of a vector $\theta$ is commonly used for $d$-way multiclass classification. It is equal to $F(\theta)$ over the unit simplex $\mathcal{C} = \{y \in \mathbb{R}^d \, : \, y \geqslant 0, \; \mathbf{1}^\top y = 1\}$. The computational cost is $O(d)$. On this set, using Gumbel noise yields the Gibbs distribution for $p_\theta$ (see below).

*Ranking.* The function returning the ranks (in descending order) of a vector $\theta \in \mathbb{R}^d$ can be written as the argmax of a linear program over the *permutahedron*, the convex hull of permutations of any vector $v$ with distinct entries $\mathcal{C} = \mathcal{P}_v = \mathrm{cvx}\{P_\sigma v \, : \, \sigma \in \Sigma_d\}$. The computational cost is $O(d \log d)$, using a sort.

*Shortest paths.* For a graph $G = (V, E)$ and positive costs over edges $c \in \mathbb{R}^E$, the problem of finding a shortest path (i.e., with minimal total cost) from vertices $s$ to $t$ can be written in our setting with $\theta = -c$ and $\mathcal{C} = \{y \in \mathbb{R}^E \, : \, y \geqslant 0, (\mathbf{1}_{\to i} - \mathbf{1}_{i \to})^\top y = \delta_{i=s} - \delta_{i=t}\}$. The computational cost is $O(|V|^2)$, using Dijkstra's algorithm.

**A generalization of Gumbel-max.** An example of this setting is well-known: when $\mathcal{Y}$ is the set of one-hot-encoding of $d$ classes, $\mathcal{C}$ is the unit simplex, and $Z$ has the Gumbel distribution [23].

In that case it is well-known that $p_\theta$ is the Gibbs distribution, proportional to $\exp(\langle y, \theta \rangle / \varepsilon)$, $F_\varepsilon(\theta)$ is the log-sum-exp function of $\theta$, and $y_\varepsilon^*(\theta)$ is the vector of softmax (or exponential weights) of the components of $\theta$. Our model is therefore a *generalization* of the Gumbel-max setting. As $F_\varepsilon$ generalizes the log-sum-exp function for Gumbel noise on the simplex, its dual $\Omega$ is a *generalization* of the negative Shannon entropy (which is the Fenchel dual of the log-sum-exp function). We show this connection, and that the perturbed maximizer can also be defined as the solution of a convex problem (by Fenchel-Rockafellar duality) in Proposition 2.1 below. The following table summarizes those parallels. Our framework generalizes these ideas, and proposes to exploit the ease of simulation of $p_\theta$ (rather than the explicit forms of Gibbs distributions) for applications in machine learning tasks.

|  | Gumbel-max | General perturbed optimizer |
|---|---|---|
| noise distribution | $Z_i$ independent Gumbel | $Z \sim \mu$, general random |
| domain $\mathcal{C}$ | unit simplex $\Delta^n$ | general polytope: $\mathrm{cvx}(\mathcal{Y})$ |
| argmax distribution | $p_{\mathsf{Gibbs},\theta} \propto \exp(\langle y, \theta \rangle / \varepsilon)$ | $p_\theta$, no closed form |
| expectation of maximum | log-sum-exp of $\theta$ | general $F_\varepsilon(\theta)$ |
| convex regularizer | Shannon negentropy | general $\Omega = (F_\varepsilon)^\star$ |

As pointed out in the remark above, the link between perturbation and regularization extends to our setting, and the perturbed maximizer can always be expressed as a regularized maximizer (see Proposition 2.1 below). This formulation, and the properties of the convex regularizer $\Omega$ yield some of the properties in Proposition 2.2 and 2.3, as well as the link with structured losses.

**Proposition 2.1.** *Let $\Omega$ be the Fenchel dual of $F_1$, with domain $\mathcal{C}$. We have that*

$$y_\varepsilon^*(\theta) = \arg\max_{y \in \mathcal{C}} \left\{ \langle y, \theta \rangle - \varepsilon\, \Omega(y) \right\}. \tag{2}$$

**Differentiation and associated loss function.** While these connections have been studied before [27, 1, 2], we provide two key new insights. First, the perturbed model allows to take derivatives with respect to the input $\theta$ of $F_\varepsilon$ and of $y_\varepsilon^*$ (Proposition 2.2). These derivatives are also easily expressed as expectations involving $F$ and $y^*$ with noisy inputs, as discussed in Section 3. In turn, this yields fast computational methods for these functions and their derivatives. Second, by the duality point of view describing $y_\varepsilon^*$ as a regularized maximizer, there exists a natural convex loss for this model that can be efficiently optimized in $\theta$, for data $y_i \in \mathcal{Y}$. We describe this formalism in Section 4, and apply it in experiments in Section 5.

**Properties of the model.** This model modifies the maximum and maximizer by perturbation. Because of the simple action of the stochastic noise , we can analyze their properties precisely.

**Proposition 2.2.** *Assume $\mathcal{C}$ is a convex polytope with non-empty interior, and $\mu$ has positive differentiable density. The perturbed model $p_\theta$ and the associated functions $F_\varepsilon$, $\Omega = (F_\varepsilon)^\star$, and $y_\varepsilon^*$ have the following properties, for $R_\mathcal{C} = \max_{y \in \mathcal{C}} \|y\|$ and $M_\mu = \mathbf{E}[\|\nabla_z \nu(Z)\|^2]^{1/2}$:*

- *$F_\varepsilon$ is strictly convex, twice differentiable, $R_\mathcal{C}$-Lipschitz-continuous and its gradient is $R_\mathcal{C} M_\mu / \varepsilon$-Lipschitz-continuous. Its dual $\Omega$ is $1/(R_\mathcal{C} M_\mu)$-strongly convex, differentiable, and Legendre-type.*

- *For all $\theta \in \mathbb{R}^d$, $y_\varepsilon^*(\theta)$ is in the interior of $\mathcal{C}$ and $y_\varepsilon^*$ is differentiable in $\theta$.*

- *Impact of $\varepsilon > 0$: we have $F_\varepsilon(\theta) = \varepsilon F_1\big(\frac{\theta}{\varepsilon}\big)$, $F_\varepsilon^*(y) = \varepsilon \Omega(y)$, $y_\varepsilon^*(\theta) = y_1^*\big(\frac{\theta}{\varepsilon}\big)$.*

We develop in further details the simple expressions for derivatives of $F_\varepsilon$ and $y_\varepsilon^*$ in Section 3. By this proposition, since $F_\varepsilon$ is strictly convex, it is nowhere locally linear, so $y_\varepsilon^*$ is nowhere locally constant. Formally, $F_\varepsilon$ is a *mirror map*, its gradient is a one-to-one mapping from $\mathbb{R}^d$ unto the interior of $\mathcal{C}$. The gradient of $\varepsilon\Omega$ is its functional inverse, by convex duality between these functions (see, e.g., surveys [55, 12] and references therein).

*Remark* 1. For these properties to hold, it is crucial that $\mathcal{C}$ has non-empty interior, i.e., that $\mathcal{Y}$ does not lie in an affine subspace of lower dimension. To adapt to cases where $\mathcal{C}$ lies in a subspace, we consider the set of inputs $\theta$ up to vectors orthogonal to $\mathcal{C}$, or represent $\mathcal{Y}$ in a lower-dimensional subspace. As an example, over the unit simplex and Gumbel noise, the log-sum-exp is not strictly convex, and in fact linear along the all-ones vector $\mathbf{1}$. In such cases, the model is only well-specified in $\theta$ up to the space orthogonal to $\mathcal{C}$, which does not affect prediction tasks.

For any positive temperature $\varepsilon$, these properties imply that there is an informative, well-defined, and nonzero gradient in $\theta$. They also imply the limiting behavior at extreme temperatures.

**Proposition 2.3.** *With the conditions of Proposition 2.2, for $\theta$ such that $y^*(\theta)$ is a unique maximum:*

*For $\varepsilon \to 0$, $F_\varepsilon(\theta) \to F(\theta)$ and $y^*_\varepsilon(\theta) \to y^*(\theta)$. For $\varepsilon \to \infty$, $y^*_\varepsilon(\theta) \to y^*_1(0) = \arg\min_{y \in \mathcal{C}} \Omega(y)$.*

*For every $\varepsilon > 0$, we have $F(\theta) - F_\varepsilon(\theta) \leqslant C\varepsilon$ and $\langle y^*(\theta), \theta \rangle - \langle y^*_\varepsilon(\theta), \theta \rangle \leqslant C'\varepsilon$, for $C, C' > 0$.*

The properties of the distributions $p_\theta$ in this model are well studied in the perturbations literature (see, e.g., [27] for a survey). They notably do not have a simple closed-form expression, but can be very easy to sample from. By the argmax definition, simulating $Y \sim p_\theta$, only requires to sample $\mu$ (e.g., Gaussian, or vector of i.i.d. Gumbel), and to solve the original optimization problem. It is the case in the applications we consider (e.g., max, ranking, shortest paths). This is in stark contrast to the Gibbs distribution, which has the opposite properties.

# 3 Differentiation of soft maximizers

As noted above, for the right noise distributions, the perturbed maximizer $y^*_\varepsilon$ is differentiable in its inputs, with non-zero Jacobian. It is based on integration by parts, not on finite differences as in [32].

**Proposition 3.1.** *[2, Lemma 1.5] For noise $Z$ with distribution $\mathrm{d}\mu(z) \propto \exp(-\nu(z))\mathrm{d}z$ and twice differentiable $\nu$, the following holds (with $J_\theta\, y^*_\varepsilon(\theta)$ the Jacobian matrix of $y^*_\varepsilon$ at $\theta$):*

$$F_\varepsilon(\theta) = \mathbf{E}[F(\theta + \varepsilon Z)], \quad y^*_\varepsilon(\theta) = \nabla_\theta F_\varepsilon(\theta) = \mathbf{E}[y^*(\theta + \varepsilon Z)] = \mathbf{E}[F(\theta + \varepsilon Z)\nabla_z\nu(Z)/\varepsilon],$$

$$J_\theta\, y^*_\varepsilon(\theta) = \mathbf{E}[y^*(\theta + \varepsilon Z)\nabla_z\nu(Z)^\top/\varepsilon] = \mathbf{E}[F(\theta + \varepsilon Z)(\nabla_z\nu(Z)\nabla_z\nu(Z)^\top - \nabla_z^2\nu(Z))/\varepsilon^2].$$

The derivatives are simple expectations. We discuss in the following subsection efficient techniques to evaluate in practice $y^*_\varepsilon(\theta)$ and its Jacobian, or to generate stochastic gradients, based on these expressions. Our method therefore provides automatically a smoothing of the function and an unbiased stochastic estimate of the corresponding Jacobian, which is always non-zero. This can be contrasted with methods relying on implicit differentiation or finite differences without smoothing, that can lead to zero Jacobians, and does not require an ad hoc solver, as would be needed for log-barrier approaches [48].

*Remark* 2. Being able to compute the perturbed maximizer and its Jacobian allows to optimize functions that depend on $\theta$ through $y^*_\varepsilon(\theta)$. This can be used to alter the costs to promote solutions with certain desired properties. Moreover, in a supervised learning setting, this allows to train models containing blocks with inputs $\theta = g_w(x)$, for some feature vector $x$, by minimizing a loss $\ell$ between the perturbed maximizer $y^*_\varepsilon(\theta)$ and the ground-truth $y$,

$$\ell(y^*_\varepsilon(\theta), y) = \ell(y^*_\varepsilon(g_w(x)), y). \tag{3}$$

For first-order methods, differentiating the above w.r.t. $w$ requires not only the usual model-dependent Jacobian $J_w g_w(x)$, but also a gradient in the first argument of the loss $\ell$. If this block is a strict discrete maximizer $y^*$, as noted above, the computational graph is broken during the backward pass: the gradient provides no information for learning. However, with our proposed modification, we have that the gradient of Eq. (3) w.r.t. $\theta$ is equal to

$$J_\theta\, y^*_\varepsilon(\theta)\, \nabla\ell(y^*_\varepsilon(\theta), y), \tag{4}$$

where $\nabla\ell$ is the gradient w.r.t. the first argument of $\ell$. Thus, the gradient can be fully backpropagated. Perturbed maximizers can therefore be used in end-to-end prediction models, for any loss $\ell$ on the perturbed maximizer. Furthermore, we describe in Section 4 a loss that can be directly optimized in $\theta$ by first-order methods. It comes with a strong algorithmic advantage, as it requires only to compute the perturbed maximizer and not its Jacobian.

**Practical implementation.** For any $\theta$, the perturbed maximizer $y^*_\varepsilon(\theta)$ is a solution of a convex optimization problem in Eq. (2), allowing computation if $\Omega$ has a simple form. More generally, by their expressions as expectations, the perturbed maximizer and its Jacobian can be approximated with Monte-Carlo methods. This only requires to efficiently sample from $\mu$, and to solve LPs over $\mathcal{C}$.

**Definition 3.1.** Given $\theta \in \mathbb{R}^d$, let $(Z^{(1)}, \ldots, Z^{(M)})$ be $M$ i.i.d. copies of $Z$ and, for $m = 1, \ldots, M$,

$$y^{(m)} = y^*(\theta + \varepsilon Z^{(m)}) = \arg\max_{y \in \mathcal{C}}\langle y, \theta + \varepsilon Z^{(m)}\rangle.$$

A *Monte-Carlo estimate* $\bar{y}_{\varepsilon,M}(\theta)$ of $y^*_\varepsilon(\theta)$ is given by $\bar{y}_{\varepsilon,M}(\theta) = \frac{1}{M}\sum_{m=1}^{M} y^{(m)}$.

Since $\mathbf{E}[y^{(m)}] = y_\varepsilon^*(\theta)$ for every $m \in \{1, \ldots, M\}$, by definition of $p_\theta$, it is an unbiased estimate of $y_\varepsilon^*(\theta)$. Note that the formulae in Proposition 3.1 give several manners to stochastically approximate $F_\varepsilon$, $y_\varepsilon^*$, and their derivatives by using $F(\theta + \varepsilon Z^{(m)})$, $y^*(\theta + \varepsilon Z^{(m)})$ and $\nabla_z \nu(Z^{(m)})$ and averages. This yields unbiased estimates for $F_\varepsilon$, $y_\varepsilon^*$, and its Jacobian. The plurality of these formulae gives the user several options for practical implementation. For both $y_\varepsilon^*$ and its Jacobian, we use the first one presented in Proposition 3.1 for our applications.

A great strength of this method is the absence of conceptual or computational overhead. Further, even though our analysis relies on the specific structure of the problem as an LP, these algorithms do not. The Monte-Carlo estimates can be obtained by using a function $y^*$ as a blackbox, without requiring knowledge of the problem or of the algorithm that solves it. For instance, for ranking, solving the LP only involves a sort.

If $y_\varepsilon^*$ or its derivatives are used in stochastic gradient descent for training in supervised learning, a full approximation of the gradients is not always necessary. Taking only $M = 1$ (or a small number) of observations is acceptable here, as the gradients are stochastic in the first place.

With parallelization and warm starts, we can alleviate the dependency in $M$ of the running time: We can independently sample the $Z^{(m)}$ and compute the $y^{(m)} = y^*(\theta + \varepsilon Z^{(m)})$ in parallel. On the other hand, starting from a solution or near-solution (such as $y^*(\theta)$) as initialization can improve running times dramatically, especially at lower temperatures. In our experiments on GPU, this led to running times almost independent of $M$ (for values up to 1000).

## 4 Perturbed model learning with Fenchel-Young losses

There is a large literature on learning parameters of a Gibbs distribution based on data $(y_i)_{i=1,\ldots,n}$, through maximization of the likelihood:

$$\bar{\ell}_n(\theta) = \frac{1}{n} \sum_{i=1}^n \log p_{\mathsf{Gibbs},\theta}(y_i) = \frac{1}{n} \sum_{i=1}^n \langle y_i, \theta \rangle - \log Z(\theta) \text{ with } \nabla_\theta \bar{\ell}_n(\theta) = \frac{1}{n} \sum_{i=1}^n y_i - \mathbf{E}_{\mathsf{Gibbs},\theta}[Y].$$
(5)

The expression of the gradient justifies the name of moment-matching procedures. The expectation of the Gibbs is however hard to evaluate in some cases. For instance, for permutation problems, it is known to be #P-hard to compute [52, 50]. This motivates its replacement by $p_\theta$ (perturb-and-MAP in this literature), and to use this method as a proxy for log-likelihood to learn the parameters [40].

We show here that this approach can be formally analyzed by the use of Fenchel-Young losses [8] in this context. It is equivalent to maximizing a term akin to Eq. (5), substituting the log-partition $Z(\theta)$ with $F_\varepsilon(\theta)$. The use of these losses also drastically improves the algorithmic aspects of the learning tasks, by the specific expression of the gradients of the loss.

**Definition 4.1.** In the perturbed model, the *Fenchel-Young* loss $L_\varepsilon(\cdot\,;y)$ is defined for $\theta \in \mathbb{R}^d$ by

$$L_\varepsilon(\theta\,;y) = F_\varepsilon(\theta) + \varepsilon\,\Omega(y) - \langle \theta, y \rangle.$$

It is nonnegative, convex in $\theta$, and minimized with value 0 if and only if $\theta$ is such that $y_\varepsilon^*(\theta) = y$. It is equal to the Bregman divergence associated to $\varepsilon\Omega$, i.e., $L_\varepsilon(\theta\,;y) = D_{\varepsilon\Omega}(y, \hat{y}_\varepsilon^*(\theta))$. As $\theta$ and $y$ interact in this loss only through a scalar product, for random $Y$ we have $\mathbf{E}[L_\varepsilon(\theta;Y)] = L_\varepsilon(\theta;\mathbf{E}[Y]) + C$, where $C$ does not depend on $\theta$. This is particularly convenient in analyzing the performance of Fenchel-Young losses in generative models. The gradient of the loss is

$$\nabla_\theta L_\varepsilon(\theta\,;y) = \nabla_\theta F_\varepsilon(\theta) - y = y_\varepsilon^*(\theta) - y\,.$$

The Fenchel-Young loss can therefore be interpreted as a loss in $\theta$ that is a function of $y_\varepsilon^*(\theta)$. Moreover, it can be optimized in $\theta$ with first-order methods simply by computing the soft maximizer, without having to compute its Jacobian. It is therefore a particular case of the situation described in Eq.(3) and (4), allowing to even bypass virtually the perturbed maximizer block in the output, and to directly optimize a loss between observation $y$ and model outputs $\theta = g_w(x)$. An interesting, yet perhaps counter-intuitive aspect of this approach is that the Fenchel–Young loss can be optimized with gradients, without being computed (which requires to also solve an optimization problem).

**Supervised and unsupervised learning.** As described in Remark 2, given observations $(x_i, y_i)_{1 \leqslant i \leqslant n} \in \mathcal{X}^n \times \mathcal{Y}^n$, we can fit a model $g_w$ such that $y_\varepsilon^*(g_w(x_i)) \approx y_i$. The Fenchel-Young loss between $g_w(x_i)$ and $y_i$ is a natural way to do so

$$L_{\varepsilon,\mathsf{emp}}(w) = \frac{1}{n} \sum_{i=1}^n L_\varepsilon(g_w(x_i)\,;y_i)\,, \text{ motivated by model } y_i = \arg\max_{y \in \mathcal{C}} \langle g_{w_0}(x_i) + \varepsilon Z^{(i)}, y \rangle.$$

Indeed, under this generative model for some $w_0$, the *population loss* $\mathbf{E}[L_{\varepsilon,\mathsf{emp}}(w)]$ is the average of terms $L_\varepsilon(g_w(x_i) ; y_\varepsilon^*(g_{w_0}(x_i)))$, up to an additive constant. The population loss is therefore minimized at $w_0$. The gradient of the empirical loss is given by

$$\nabla_w L_{\varepsilon,\mathsf{emp}}(w) = \tfrac{1}{n}\sum_{i=1}^n J_w\, g_w(x_i)\cdot (y_\varepsilon^*(g_w(x_i)) - y_i)\,.$$

Each term in the sum, gradient of the loss for a single observation, is therefore a stochastic gradient for $L_{\varepsilon,\mathsf{emp}}$ (w.r.t. $i$ uniform in $[n]$) or for $L_{\varepsilon,\mathsf{pop}}$ (w.r.t. to a random $y_i$ from $p_{g_{w_0}(x_i)}$).

The methods we described to stochastically approximate the gradient are particularly adapted here. Indeed, following [49], given an observation $y_i$ and a current value $\theta_i = g_w(x_i)$, a *doubly stochastic* version of the gradient $\nabla_w L_\varepsilon(g_w(x_i) ; y_i)$ is obtained by

$$\bar{\gamma}_{i,M}(w) = J_w\, g_w(x_i)\big(\tfrac{1}{M}\sum_{m=1}^M y^*\big(g_w(x_i) + \varepsilon Z^{(m)}\big) - y_i\big)\,. \tag{6}$$

This can also be used with a procedure where batches of data points are used to compute approximate gradients, where the number of artificial samples $M$ and the batch size can be chosen separately.

This can be extended to an unsupervised setting, where observations $(y_i)_{1\leqslant i\leqslant n} \in \mathcal{Y}^n$ are fitted with a model $p_\theta$, motivated by a generative model where $y_i = \arg\max_{y\in\mathcal{C}} \langle \theta_0 + \varepsilon Z_i, y\rangle$, that is $y_i \sim p_{\theta_0}(y)$, for some unknown $\theta_0$. We have a natural *empirical* $\bar{L}_n$ and *population* loss $L_{\theta_0}$:

$$\bar{L}_{\varepsilon,n}(\theta) = \tfrac{1}{n}\sum_{i=1}^n L_\varepsilon(\theta; y_i) = L_\varepsilon(\theta; \bar{Y}_n) + C(Y)\,,\quad L_{\varepsilon,\theta_0}(\theta) = \mathbf{E}[\bar{L}_{\varepsilon,n}(\theta)] = L_\varepsilon(\theta; y_\varepsilon^*(\theta_0)) + C(\theta_0)\,.$$

Their gradients are given by

$$\nabla_\theta \bar{L}_{\varepsilon,n}(\theta) = \nabla_\theta F_\varepsilon(\theta) - \bar{Y}_n = y_\varepsilon^*(\theta) - \bar{Y}_n\,,\quad \text{and}\quad \nabla_\theta L_{\varepsilon,\theta_0}(\theta) = y_\varepsilon^*(\theta) - y_\varepsilon^*(\theta_0)\,.$$

The empirical loss is minimized for $\hat{\theta}_n$ such that $y_\varepsilon^*(\hat{\theta}_n) = \bar{Y}_n$ and the population loss when $y_\varepsilon^*(\theta) = y_\varepsilon^*(\theta_0)$. As a consequence, the whole battery of statistical results, from asymptotic to non-asymptotic, can be leveraged, and we present the simplest one (asymptotic normality).

**Proposition 4.1.** *When $n$ goes to $\infty$, with the assumptions of Proposition 2.2 on the model, we have*

$$\sqrt{n}(\hat{\theta}_n - \theta_0) \to \mathcal{N}\big(0, \big(\nabla_\theta^2 F_\varepsilon(\theta_0)\big)^{-1}\Sigma_Y\big(\nabla_\theta^2 F_\varepsilon(\theta_0)\big)^{-1}\big)\,,$$

*in distribution, where $\Sigma_Y$ is the covariance of $Y \sim p_\theta$.*

## 5    Experiments

We demonstrate the usefulness of perturbed maximizers in a supervised learning setting, as described in Section 4. We focus on a classification task and on two structured prediction tasks, label ranking and learning to predict shortest paths. Since we focus on the prediction task, the issues raised in Remark 1 do not apply. When learning with the Fenchel-Young losses, we simulate doubly stochastic gradients $\nabla_w L_\varepsilon(g_w(x_i) ; y_i)$ of the empirical loss with $M$ artificial perturbations (see Equation 6).

We will open-source a Python package allowing to turn any black-box solver into a differentiable function, in just a few lines of code. Full details of the experiments are included in Appendix C.

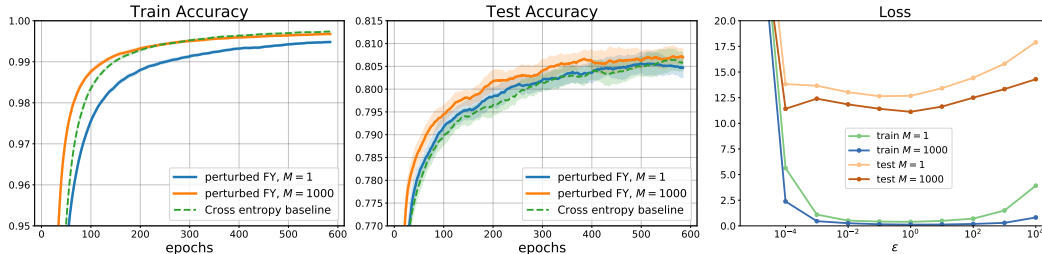

Figure 2: **Left.** Accuracy in training, using the perturbed FY loss, or cross entropy baseline. **Center.** Test accuracy for these methods. **Right.** Impact of the parameter $\varepsilon$ on test and train squared loss.

## 5.1 Perturbed max

We use the perturbed argmax with Gaussian noise in an image classification task on the CIFAR-10 dataset. This serves two purposes: showing that we perform as well as the cross entropy loss, in a case where a soft max can be easily computed, and exhibiting the impact of the algorithmic parameters. We train a vanilla-CNN with 10 network outputs that are the entries of $\theta$, we minimize the Fenchel-Young loss between $\theta_i = g_w(x_i)$ and $y_i$, with different temperatures $\varepsilon$ and number of perturbations $M$. We observe competitive performance compared to standard losses as baselines (Fig. 2, left and center).

We analyze the impact of the algorithmic parameters on optimization and generalization abilities. We exhibit the final loss and accuracy for different number of perturbations in the doubly stochastic gradient ($M = 1, 1000$). We highlight the importance of the temperature parameter $\varepsilon$ on the algorithm (see Figure 2, right). Very high or low temperatures degrade the ability to fit to training and to generalize to test data, by lack of smoothing or loss of information about $\theta$. We also observe that our framework is very robust to the choice of $\varepsilon$, demonstrating its adaptivity.

## 5.2 Perturbed label ranking

We consider label ranking tasks, where each $y_i$ is a label permutation for features $x_i$. We minimize the weights of an affine model $g_w$ (i.e., $\theta_i = g_w(x_i)$) using our perturbed Fenchel-Young loss, a simple squared loss and the recently-proposed method of gradient estimation Vlastelica et al. [54]. Note that our loss is convex in $\theta$ and enjoys unbiased gradients, while [54] uses a non-convex loss with gradient proxies. We use the same 21 datasets as in [28, 14]. We report Spearman's correlation (higher is better) in Figure 3. Results are averaged over 10-fold CV and parameters tuned by 5-fold CV. We find that the FY loss performs better or similarly (within a 5% range) on 76 % and 90 % of the datasets, respectively. Detailed experimental setup and results are given in Appendix C.2.

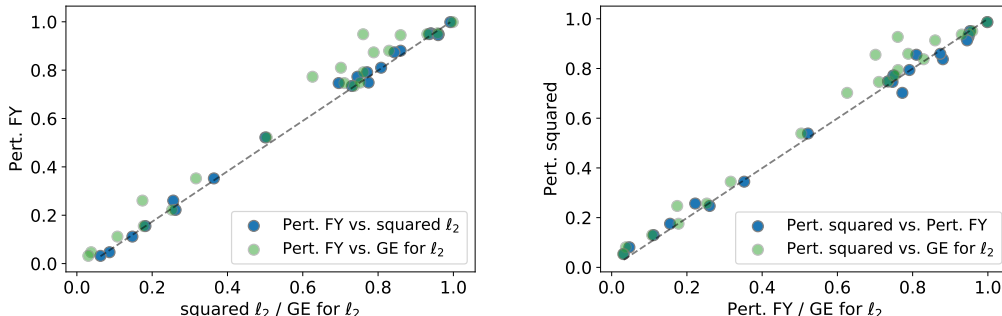

Figure 3: Comparison of Spearman correlations on 21 datasets **Left.** Our proposed perturbed Fenchel-Young (FY) loss, a squared loss and the gradient estimation (GE) for $\ell_2$ loss of [54] **Right.** Our proposed perturbed squared $\ell_2$ loss, our perturbed FY, and GE for $\ell_2$. Points above the diagonal are datasets where our loss on the $y$-axis performs better.

To better understand the complexity of this task, we also created a range of artificial datasets where 100 labels are generated by taking a randomly perturbed $y_i = \arg\max_y \langle x_i^\top w_0 + \sigma Z_i, y \rangle$, in dimension 50, for different values of $\sigma$. We minimize the same losses as before in $w$. For almost correct labels ($\sigma \approx 0$), our method accurately generalizes to the test data (see Figure 3, and Figure 7 in Appendix C for other metrics). We observe that the Fenchel-Young loss performs as well or better than the other losses, particularly in terms of robustness to the noise. All details are included in Appendix C.2.

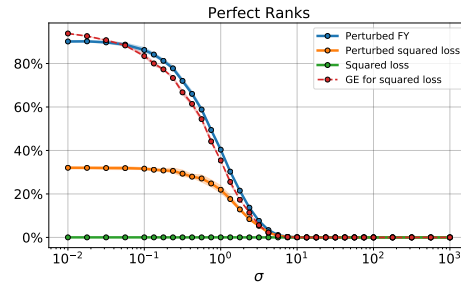

Figure 4: Average number of instances with exactly correct ranks for all 100 labels, for different values of $\sigma$, for four methods.

## 5.3 Perturbed shortest path

We replicate the experiment of Vlastelica et al. [54], aiming to learn the travel costs in graphs based on features, given examples of shortest path solutions (see Figure 5). We use a dataset of 10,000 RGB images of size $96 \times 96$ illustrating Warcraft terrains of $12 \times 12$ 2D grid networks. The responses $y_i$ are a shortest path between the top-left and bottom-right corners, for costs hidden to the network, corresponding to the terrain type. They are $12 \times 12$ binary matrices representing the vertices along the shortest path.

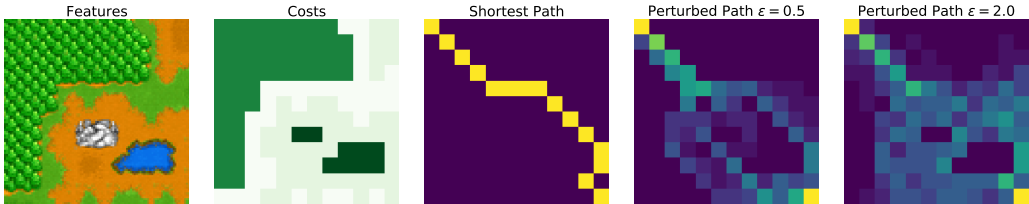

Figure 5: In the shortest path experiment, training features are images. Shortest paths are computed based on terrain costs, hidden to the network. Training responses are shortest paths based on this cost.

Following [54], we train a network whose first five layers are those of ResNet18 for the Fenchel-Young loss between the predicted costs $\theta_i = g_w(x_i)$ and the shortest path $y_i$. We optimize over 50 epochs with batches of size 70, temperature $\varepsilon = 1$ and $M = 1$ (single perturbation). We are able, only after a few epochs, to generalize very well, and to accurately predict the shortest path on the test data. We compare our method to two baselines, from [54]: training the same network with their proposed gradient estimation, and with a squared loss. We show two metrics: perfect accuracy percentage and cost ratio to optimal path (see Figure 6); full implementation details are in Appendix C.3.

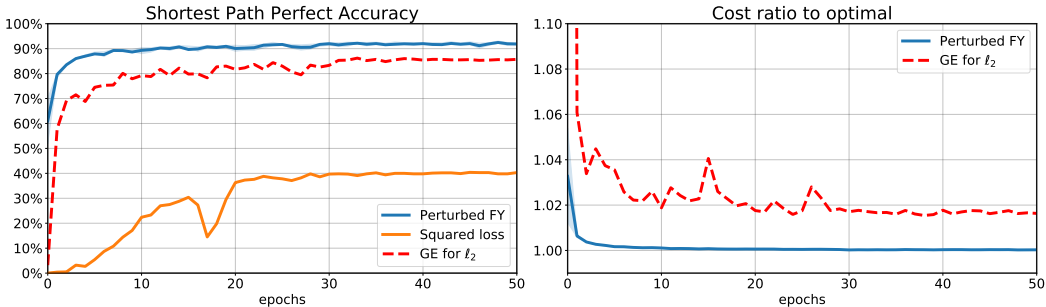

Figure 6: Accuracy of the predicted path for several methods during training. **Left.** Percentage of test instances where the predicted path is optimal. **Right.** Ratio of costs between the predicted path and the actual shortest path – without the squared loss baseline as it does not yield valid paths.

## 6 Conclusion

Despite a large body of work on perturbations techniques for machine learning, most existing works focused on approximating sampling, log-partitions and expectations under the Gibbs distribution. Together with novel theoretical insights, we propose to use a general perturbation framework to differentiate through, not only a max, but also an argmax, without ad-hoc modification of the underlying solver. In addition, by defining an equivalent regularizer $\Omega$, we show how to construct Fenchel-Young losses and propose a doubly stochastic scheme, enabling learning in various tasks, and validate on experiments its ease of application.

## Broader impact

This submission focuses on foundational work, with application to general machine learning techniques. These techniques expend the range of operations that can be used in end-to-end differentiable systems, by allowing to incorporate optimizers in ML pipelines. There are no foreseeable societal consequences that are specifically related to these methods, beyond those of the field in general.

## Acknowledgments and Disclosure of Funding

FB's work was funded in part by the French government under management of Agence Nationale de la Recherche as part of the "Investissements d'avenir" program, reference ANR-19-P3IA-0001 (PRAIRIE 3IA Institute). FB also acknowledges support from the European Research Council (grant SEQUOIA 724063).

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
