[Supplementary Material]

# A Proofs of technical results

*Proof of Proposition 2.1.* The function $\varepsilon\Omega$ is the Fenchel dual of $F_\varepsilon$ (see Proposition 2.2, impact of the temperature), and is defined on $\mathcal{C}$. As such, as in [1], we have that

$$F_\varepsilon(\theta) = \sup_{y \in \mathcal{C}} \left\{ \langle \theta, y \rangle - \varepsilon\Omega(y) \right\}.$$

It is maximized at $\nabla_\theta F_\varepsilon(\theta) = y_\varepsilon^*(\theta)$, by Fenchel-Rockaffelar duality [see, e.g. 55, Appendix A]. $\square$

*Proof of Proposition 2.2.* The proof of these properties makes use of the notion of the *normal fan* of $\mathcal{C}$. It is the set of all *normal cones* to all faces of the polytope $\mathcal{C}$ [43]. For each face, such a cone is the set of vectors in $\mathbb{R}^d$ such that the linear program on $\mathcal{C}$ with this vector as cost is maximized on this face. They form a partition of $\mathbb{R}^d$, and these cones are full dimensional if and only if they are associated to a vertex of $\mathcal{C}$. These vertices are a subset $\mathcal{E}$ of $\mathcal{Y}$, corresponding to extreme points of $\mathcal{C}$.

As a consequence of this normal cone structure, since $\mu$ has a positive density, it assigns positive mass to sets if and only if they have non-empty interior, so for any $\theta \in \mathbb{R}^d$, and any $\varepsilon > 0$, $p_\theta(y) > 0$ if and only if $y \in \mathcal{E}$. In most applications, $\mathcal{E} = \mathcal{Y}$ to begin with (all $y$ are potential maximizer for some vector of costs, otherwise they are not included in the set), and all points in $\mathcal{Y}$ have positive mass.

**Properties of $F_\varepsilon$**

- $F_\varepsilon$ is strictly convex

The function $F$ is convex, as a maximum of convex (linear) functions. By definition of $F_\varepsilon$, for every $\lambda \in [0,1]$ and $\theta, \theta' \in \mathbb{R}^d$, for $\theta_\lambda = \lambda\theta + (1-\lambda)\theta'$ we have

$$\lambda F_\varepsilon(\theta) + (1-\lambda)F_\varepsilon(\theta') = \mathbf{E}[\lambda F(\theta + \varepsilon Z) + (1-\lambda)F(\theta' + \varepsilon Z)] \leqslant \mathbf{E}[F(\lambda\theta + (1-\lambda)\theta' + \varepsilon Z)] = F_\varepsilon(\theta_\lambda).$$

The inequality holds with equality if and only if it holds within the expectation for almost all $z$ since the distribution of $Z$ is positive on $\mathbb{R}^d$. If the function $F_\varepsilon$ is not strictly convex, there exists therefore $\theta$ and $\theta'$ such that

$$\lambda F(\theta + \varepsilon z) + (1-\lambda)F(\theta' + \varepsilon z) = F(\lambda\theta + (1-\lambda)\theta' + \varepsilon z)$$

for all $\lambda \in [0,1]$, for almost all $z \in \mathbb{R}^d$. In this case, $F$ is linear on the segment $[\theta + \varepsilon z, \theta' + \varepsilon z]$ for almost all $z \in \mathbb{R}^d$.

If $\theta - \theta'$ is contained in the boundary between the normal cones to $y_1$ and $y_2$, for all distinct $y_1, y_2 \in \mathcal{E}$, we have $\langle y_1 - y_2, \theta - \theta' \rangle = 0$ for all such pairs of $y$, so $\theta$ is orthogonal to the span of all the pairwise differences of $y$. However, since $\mathcal{C}$ has no empty interior, it is not contained in a strict affine subspace of $\mathbb{R}^d$ so $\theta - \theta' = 0$. As a consequence, for distinct $\theta$ and $\theta'$, there exists $z \in \mathbb{R}^d$ such that $\theta + \varepsilon z$ and $\theta + \varepsilon z$ are in the interior of two normal cones to different $y \in \mathcal{E}$. As a consequence, the same holds under perturbations of $z$ in a small enough ball of $\mathbb{R}^d$, so $F$ cannot be linear on almost all segments $[\theta + \varepsilon z, \theta' + \varepsilon z]$, and $F_\varepsilon$ is strictly convex.

- $F_\varepsilon$ is twice differentiable, as a direct consequence of Proposition 3.1.

- $F_\varepsilon$ is $R_\mathcal{C}$-Lipschitz

$F$ is the maximum of finitely many functions that are $R_\mathcal{C}$-Lipschitz. It therefore also satisfies this property. $F_\varepsilon$ is an expectation of such functions, therefore it satisfies the same property.

- $F_\varepsilon$ is $R_\mathcal{C}M_\mu/\varepsilon$-gradient Lipschitz.

We have, by Proposition 3.1, for $\theta$ and $\theta'$ in $\mathbb{R}^d$

$$\nabla_\theta F_\varepsilon(\theta) - \nabla_\theta F_\varepsilon(\theta') = \mathbf{E}[(F(\theta + \varepsilon Z) - F(\theta' + \varepsilon Z))\nabla_z \nu(Z)/\varepsilon].$$

As a consequence, by the Cauchy–Schwarz inequality, and Lipschitz property of $F$, it holds that

$$\|\nabla_\theta F_\varepsilon(\theta) - \nabla_\theta F_\varepsilon(\theta')\| \leqslant \mathbf{E}[\|F(\theta + \varepsilon Z) - F(\theta' + \varepsilon Z)\|^2]^{1/2}\mathbf{E}[\|\nabla_z \nu(Z)\|^2/\varepsilon^2]^{1/2}$$

$$\leqslant R_\mathcal{C}\|\theta - \theta'\|\mathbf{E}[\|\nabla_z \nu(Z)\|^2]^{1/2}/\varepsilon = (R_\mathcal{C}M_\mu/\varepsilon)\|\theta - \theta'\|.$$

**Properties of $\Omega$**

The function $\varepsilon\Omega$ is the Fenchel dual of $F_\varepsilon$, which is strictly convex and $R_\mathcal{C} M_\mu/\varepsilon$ smooth. As a consequence, $\Omega$ is differentiable on the image of $y_\varepsilon^*$ – the interior of $\mathcal{C}$ – and it is $1/R_\mathcal{C} M_\mu$-strongly convex.

- Legendre type property

The regularization function $\Omega$ is differentiable on the interior. If there is a point $y$ of its boundary such that $\nabla_y \Omega$ does not diverge when approaching $y$, then taking $\theta$ such that $\theta - \varepsilon\nabla_y\Omega(y) \in \mathcal{N}_\mathcal{C}(y)$ (where $\mathcal{N}_\mathcal{C}(y)$ is the normal cone to $\mathcal{C}$ at $y$), then $y_\varepsilon^*(\theta) = y$. However, $y_\varepsilon^*$ takes image in the interior of $\mathcal{C}$ (see immediately below), leading to a contradiction.

**Properties of $y_\varepsilon^*$**

- The perturbed maximizer is in the interior of $\mathcal{C}$

Since the distribution of $Z$ has positive density, the probability that $\theta + \varepsilon Z \in \mathcal{N}_\mathcal{C}(y)$ (i.e. $p_\theta(y)$) is positive for all $y \in \mathcal{E}$. As a consequence, since

$$y_\varepsilon^*(\theta) = \sum_{y \in \mathcal{E}} y\, p_\theta(y)\,,$$

with all positive weights $p_\theta(y)$, $y_\varepsilon^*$ is in the interior of the convex hull $\mathcal{C}$ of $\mathcal{E}$.

- The function $y_\varepsilon^*$ is differentiable, by twice differentiability of $F_\varepsilon$, by Proposition 3.1.

**Influence of temperature parameter** $\varepsilon > 0$

We have for all $\theta$

$$F_\varepsilon(\theta) = \mathbf{E}[\max_{y \in \mathcal{C}}\langle y, \theta + \varepsilon Z\rangle] = \varepsilon\mathbf{E}[\max_{y \in \mathcal{C}}\langle y/\varepsilon, \theta + Z\rangle] = \varepsilon F_1(\theta/\varepsilon)\,.$$

As a consequence

$$(F_\varepsilon)^*(y) = \max_{z \in \mathbb{R}^d}\{\langle y, z\rangle - F_\varepsilon(z)\} = \varepsilon\max_{z \in \mathbb{R}^d}\{\langle y, z/\varepsilon\rangle - \varepsilon F_1(z/\varepsilon)\} = \varepsilon(F_1)^*(y) = \varepsilon\,\Omega(y)\,.$$

Since $y_\varepsilon^*(\theta) = \nabla_\theta F_\varepsilon(\theta)$, and since $F_\varepsilon(\theta) = \varepsilon F_1(\theta/\varepsilon)$, we have $y_\varepsilon^*(\theta) = y_1^*(\theta/\varepsilon)$. $\qquad\square$

*Proof of Proposition 2.3.* We recall that we assume that $\theta$ yields a unique maximum to the linear program on $\mathcal{C}$. This is true almost everywhere, and assumed here for simplicity of the results. We discuss briefly at the end of this proof how this can be painlessly extended to the more general case.

**Limit at low temperatures** $(\varepsilon \to 0)$

Since $F$ is convex (see proof of Proposition 2.2), so by Jensen's inequality

$$F\big(\mathbf{E}[\theta + \varepsilon Z]\big) \leqslant \mathbf{E}\big[F(\theta + \varepsilon Z)\big]$$
$$F(\theta) \leqslant F_\varepsilon(\theta)\,.$$

Further, we have for all $Z \in \mathbb{R}^d$

$$\max_{y \in \mathcal{C}}\langle\theta + \varepsilon Z, y\rangle \leqslant \max_{y \in \mathcal{C}}\langle\theta, y\rangle + \varepsilon\max_{y' \in \mathcal{C}}\langle Z, y'\rangle$$

Taking expectations on both sides yields that

$$F_\varepsilon(\theta) \leqslant F(\theta) + \varepsilon F_1(\theta)\,.$$

As a consequence, when $\varepsilon \to 0$, combining these two inequalities yields that $F_\varepsilon(\theta) \to F(\theta)$.

Regarding the behavior of the perturbed maximizer $y_\varepsilon^*(\theta)$, we follow the arguments of [41, Proposition 4.1]. By Proposition 2.1 and the definition of $y^*(\theta)$, we have

$$0 \leqslant \langle y^*(\theta), \theta\rangle - \langle y_\varepsilon^*(\theta), \theta\rangle \leqslant \varepsilon\big[\Omega\big(y^*(\theta)\big) - \Omega\big(y_\varepsilon^*(\theta)\big)\big]$$

Since $\Omega$ is continuous, it is bounded on $\mathcal{C}$, and the right hand term above is bounded by $C\varepsilon$, for some $\varepsilon > 0$. As a consequence, when $\varepsilon \to 0$, $\langle y_\varepsilon^*(\theta), \theta\rangle \to \langle y^*(\theta), \theta\rangle$. For any sequence $\varepsilon_n \to 0$, the sequence $y_n = y_{\varepsilon_n}^*(\theta)$ is in a compact $\mathcal{C}$. Therefore, it has a subsequence $y_{\varphi(n)}$ that converges

to some limit $y_\infty \in \mathcal{C}$. However, since $\langle y^*_{\varphi(n)}, \theta \rangle \to \langle y^*(\theta), \theta \rangle$, we have $\langle y_\infty, \theta \rangle = \langle y^*(\theta), \theta \rangle$, by continuity. Since $y^*(\theta)$ is a unique maximizer, $y_\infty = y^*(\theta)$. As a consequence, all convergent subsequences of $y_n$ converge to the same limit $y^*(\theta)$: it is the unique accumulation point of this sequence. It follows directly that $y_n$ converges to $y^*(\theta)$, as it lives in a compact set, which yields the desired result.

**Limit at high temperatures** By Proposition 2.2, $y^*_\varepsilon(\theta) = y_1(\theta/\varepsilon)$, so the desired result follows by continuity of the perturbed maximizer.

**Nonasymptotic inequalities.** These inequalities follow directly from those proved to establish limits at low temperatures.

If $\theta$ is such that the maximizer is not unique (which occurs only on a set of measure 0), the only result affected is the convergence of $y^*_\varepsilon(\theta)$ when $\theta \to 0$. Following the same proof of [41, Proposition 4.1], it can be shown to converge to the minimizer of $\Omega$ over the set of maximizer. This point is always unique, as the minimizer of a strongly convex function over a convex set. $\qquad\square$

*Proof of Proposition 4.1.* We follow the classical proofs in M-estimation [see, e.g. 53, Section 5.3]. First, the estimator is consistent as a virtue of the continuous map between $\mathbb{R}^d$ and $\text{int}(\mathcal{C})$. For $n$ large enough $\bar{Y}_n \in \text{int}(\mathcal{C})$, since the probability of each extreme point of $\mathcal{C}$ is positive. By definition of the estimator and stationarity condition for $\hat{\theta}_n$, we have in these conditions

$$\nabla_\theta F_\varepsilon(\hat{\theta}_n) = \bar{Y}_n, \ \nabla_\theta F_\varepsilon(\theta_0) = y^*_\varepsilon(\theta_0).$$

By the law of large numbers, $\bar{Y}_n$ converges to its expectation $y^*_\varepsilon(\theta_0)$ a.s. Since $\varepsilon \nabla_y \Omega$, the inverse of $\nabla_\theta F_\varepsilon$, is also continuous (by the fact that $\Omega$ is convex smooth), we have that $\hat{\theta}_n$ converges to $\theta_0$ a.s.

We write the first order conditions for $\bar{L}_{\varepsilon,n}$ at $\hat{\theta}_n$ and the Taylor expansion with Lagrange remainder for all coordinates, one by one

$$0 = \nabla_\theta \bar{L}_{\varepsilon,n}(\hat{\theta}_n) = \nabla_\theta \bar{L}_{\varepsilon,n}(\theta_0) + A_n(\hat{\theta}_n - \theta_0), \tag{7}$$

where $A$ is such that, for all coordinates $i \in [d]$

$$A_i = (\nabla^2_\theta \bar{L}_{\varepsilon,n}(\bar{\theta}^{(i)}))_i$$

for some $\bar{\theta}^{(i)} \in [\hat{\theta}_n, \theta_0]$. We note here that since the estimator is not necessarily in dimension 1, $A_n$ cannot be written directly as $\nabla^2_\theta \bar{L}_{\varepsilon,n}(\bar{\theta})$ for some $\bar{\theta} \in [\hat{\theta}_n, \theta_0]$, since the Taylor expansion with Lagrange remainder is not true in its multivariate form. However, doing it coordinate-by-coordinate as here allows to circumvent this issue.

We have that $\nabla^2 \bar{L}_{\varepsilon,n} = \nabla^2 F_\varepsilon$. Since $\hat{\theta}_n \to \theta_0$ a.s. we have that $\bar{\theta}^{(i)} \to \theta_0$ for all $i \in [d]$, so $A_n \to \nabla^2 F_\varepsilon(\theta_0)$ a.s. Rearranging terms in Eq. (7), we have

$$\sqrt{n}(\hat{\theta}_n - \theta_0) = -A_n^{-1} \cdot \sqrt{n} \nabla_\theta \bar{L}_{\varepsilon,n}(\theta_0)$$
$$= -A_n^{-1} \cdot \sqrt{n}(\bar{Y}_n - y^*_\varepsilon(\theta_0))$$

By the central limit theorem, $\sqrt{n}(\bar{Y}_n - y^*_\varepsilon(\theta_0)) \to \mathcal{N}(0, \Sigma_Y)$ in distribution. As a consequence, by convergence of $A_n$ and Slutsky's lemma, we have the convergence in distribution

$$\sqrt{n}(\hat{\theta}_n - \theta_0) \to \mathcal{N}\left(0, \left(\nabla^2_\theta F_\varepsilon(\theta_0)\right)^{-1} \Sigma_Y \left(\nabla^2_\theta F_\varepsilon(\theta_0)\right)^{-1}\right).$$

$\qquad\square$

# B Examples of discrete decision problems as linear programs

Our method applies seamlessly to all decision problems over discrete sets. Indeed, any problem of the form $\max_{y \in \mathcal{Y}} s(y)$, for some score function $s : \mathcal{Y} \to \mathbb{R}$, can at least be written in the form

$$\max_{x \in \Delta^{|\mathcal{Y}|}} \langle x, s \rangle ,$$

by representing $\mathcal{Y}$ as the vertices of the unit simplex in $\mathbb{R}^{|\mathcal{Y}|}$. However, for most interesting decision problem that can actually be solved in practice, the score function takes a simpler form $s(y) = \langle y, \theta \rangle$, for some representation of $y \in \mathbb{R}^d$ and some $\theta$. We give here a non-exhaustive list of examples of interesting problems of this type.

**Maximum.** The max function from $\mathbb{R}^d$ to $\mathbb{R}$, that returns the largest among the $d$ entries of a vector $\theta$ is ubiquitous in machine learning, the hallmark of any classification task. It is equal to $F(\theta)$ over the standard unit simplex.

$$F(\theta) = \max_{i \in [d]} \theta_i , \quad \mathcal{C} = \{ y \in \mathbb{R}^d : y \geqslant 0 , \; \mathbf{1}^\top y = 1 \} .$$

On this set, using Gumbel noise yields the log-sum-exp for $F_\varepsilon$, the Gibbs distribution for $p_\theta$, and the softmax for $y_\varepsilon^*$. Using other noise distributions for $Z$ will change the model.

**Top $k$.** The function from $\mathbb{R}^d$ to $\mathbb{R}$ that returns the sum of the $k$ largest entries of a vector $\theta$ is also commonly used. It fits our framework over the set

$$\mathcal{C} = \{ y \in \mathbb{R}^d : 0 \leqslant y \leqslant 1 , \; \mathbf{1}^\top y = k \} .$$

**Ranking.** The function returning the ranks (in descending order) of a vector $\theta \in \mathbb{R}^d$ can be written as the argmax of a linear program over the *permutahedron*, the convex hull of permutations of any vector $v$ with distinct entries

$$\mathcal{C} = \mathcal{P}_v = \mathrm{cvx}\{ P_\sigma v : \sigma \in \Sigma_d \} .$$

Using different reference vectors $v$ yield different perturbed operations, and $v = (1, 2, \ldots, d)$ is commonly used.

**Shortest paths.** For a graph $G = (V, E)$ and positive costs over edges $c \in \mathbb{R}^E$, the problem of finding a shortest path (i.e. with minimal total cost) from vertices $s$ to $t$ can be written in our setting with $\theta = -c$ and

$$\mathcal{C} = \{ y \in \mathbb{R}^E : y \geqslant 0 , (\mathbf{1}_{\to i} - \mathbf{1}_{i \to})^\top y = \delta_{i=s} - \delta_{i=t} \} .$$

**Assignment.** The linear assignment problem, and more generally the optimal transport problem, can also be written as a linear program. In the case of the assignment problem, it is the *Birkhoff polytope* of doubly-stochastic matrices, whose extreme points are the permutation matrices

$$\mathcal{C} = \{ Y \in \mathbb{R}^{d' \times d'} : Y_{ij} \geqslant 0 , \; \mathbf{1}^\top Y = \mathbf{1}^\top , \quad Y\mathbf{1} = \mathbf{1} \} .$$

There is a large literature on regularization of this problem, with entropic penalty [17]. This is one of the rare cases where the regularized version of the problem is actually computationally lighter, in stark contrast with the general case in our setting.

**Combinatorial problems.** Many other problems, such in combinatorial optimization can be formulated exactly (e.g. minimum spanning tree, maximum flow), or approximately via convex relaxations (e.g. traveling salesman problem, knapsack), via relaxations in linear programs. Differentiable versions of these exact or approximate solutions can therefore be obtained via perturbation methods.

**Relaxations with atomic norms** A wide variety of high-dimensional statistical learning problems can be tackled by regularization via atomic, or otherwise sparsity-inducing norms [13, 6]. Our framework also allows us to consider versions of these estimators that are differentiable in their inputs.

# C   Experimental details

## C.1   Perturbed maximum

In the experiment on perturbed maximum for classification on CIFAR-10, we train a vanilla-CNN made of 4 convolutional and 2 fully connected layers for 600 epochs with batches of size 32.

We train by minimizing two losses in the weights $w$ of the network function $g_w$, fitting the outputs $\theta_i = g_w(x_i)$ to labels $y_i$

– Perturbed Fenchel-Young (proposed): our proposed Fenchel-Young loss (see Definition 4.1),

$$L_\varepsilon(g_w(x_i); y_i)\,,$$

– Cross entropy loss, for a soft max layer $s_\varepsilon$ and an entrywise log

$$H(g_w(x_i); y_i) = \langle y_i, \log(s_\varepsilon(g_w(x_i)))\rangle\,.$$

## C.2   Perturbed label ranking

In this experiment, we consider label ranking tasks, where each $y_i$ is a ground-truth label permutation for features $x_i$. We minimize the weights of an affine model $g_w$ (i.e., $\theta_i = g_w(x_i)$) using the following losses:

– Perturbed Fenchel-Young (proposed): our proposed Fenchel-Young loss (see Definition 4.1),
– Perturbed + Squared loss (proposed): $\frac{1}{2}\|y_i - y_\varepsilon^*(g_w(x_i))\|^2$, where gradients can be computed using Proposition 3.1 and the chain rule,
– Squared loss: $\frac{1}{2}\|y_i - g_w(x_i)\|^2$,
– Gradient proxy: $\frac{1}{2}\|y_i - y^*(g_w(x_i))\|^2$, where we use the gradient **proxy** of Vlastelica et al. [54], re-implemented for the experiments on ranking.

We use the same 21 datasets as in [28, 14]. We use $M = 10$ in the gradient computations for this experiment. Detailed results are given in Table 1 and Table 2.

For the experiment on artificial datasets, we use the same setup as above, with a linear model instead of affine. The ground-truth vector $w_0$ is obtained by uniform sampling in $\{-1, 1\}^d$, and the $x_i$ are standard isotropic normal. In the experiments presented here, we optimize over 2000 epochs, with a batch size of 32: very good results are obtained even with a smaller number of epochs, but we increased it artificially to better evaluate numerically the *final* predictive performance of all methods (see Figure 8). In the main text, we present in Figure 4 the metric of perfect rank accuracy over one run of simulations. We present in Figure 7 the same metric, as well as the metric of partial rank accuracy (i.e. the proportion of correctly ordered labels), for completeness, averaged over three runs of the dataset. To further illustrate these results, we include in Figure 8, for two fixed values of the noise level, how these metrics evolve through training.

## C.3   Perturbed shortest path

In this experiment, we have followed the setup of Vlastelica et al. [54], to obtain comparable results. We have replicated the network that they use based on Resnet18, and followed their optimization procedure, using Adam with the same learning rate schedule, changing at epochs 30 and 40 out of 50. We also included the baseline that they used, based on training the same network without an optimizer layer. These results are obtained by using the implementation code that they provide.

We minimize the weights of this model for our proposed Fenchel-Young loss (see Definition 4.1).

The perfect accuracy metric measures the percentage of test instances for which an exactly optimal path is recovered, and the cost ratio to optimal metric measures the ratio between the total cost of the path proposed by taking the shortest path for proposed costs $\theta_i = g_w(x_i)$ (after training) to the total cost of the path with true costs (see Figure 6).

Table 1: Spearman correlation on 21 datasets averaged by 10-fold cross-validation. The learning rate is chosen from $(10^{-3}, 10^{-2}, 10^{-1})$ by grid search over 5-fold cross-validation.

| Dataset | Perturbed FY | Perturbed + Squared loss | Squared loss | Gradient proxy [54] |
|---|---|---|---|---|
| authorship | $0.95 \pm 0.01$ | $0.28 \pm 0.17$ | $\mathbf{0.96} \pm 0.01$ | $0.76 \pm 0.04$ |
| bodyfat | $0.35 \pm 0.07$ | $0.23 \pm 0.09$ | $\mathbf{0.36} \pm 0.08$ | $0.32 \pm 0.07$ |
| calhousing | $\mathbf{0.26} \pm 0.02$ | $0.13 \pm 0.07$ | $\mathbf{0.26} \pm 0.01$ | $0.17 \pm 0.06$ |
| cold | $0.05 \pm 0.04$ | $0.00 \pm 0.04$ | $\mathbf{0.09} \pm 0.04$ | $0.04 \pm 0.04$ |
| cpu-small | $\mathbf{0.52} \pm 0.01$ | $0.44 \pm 0.05$ | $0.50 \pm 0.01$ | $0.50 \pm 0.01$ |
| diau | $0.22 \pm 0.03$ | $0.12 \pm 0.05$ | $\mathbf{0.26} \pm 0.03$ | $0.25 \pm 0.03$ |
| dtt | $0.11 \pm 0.04$ | $0.03 \pm 0.06$ | $\mathbf{0.15} \pm 0.04$ | $0.11 \pm 0.04$ |
| elevators | $\mathbf{0.79} \pm 0.01$ | $0.67 \pm 0.05$ | $0.77 \pm 0.01$ | $0.76 \pm 0.02$ |
| fried | $\mathbf{1.00} \pm 0.00$ | $0.82 \pm 0.10$ | $0.99 \pm 0.00$ | $\mathbf{1.00} \pm 0.00$ |
| glass | $\mathbf{0.88} \pm 0.05$ | $0.81 \pm 0.09$ | $0.86 \pm 0.05$ | $0.83 \pm 0.05$ |
| heat | $0.03 \pm 0.03$ | $0.01 \pm 0.03$ | $\mathbf{0.06} \pm 0.02$ | $0.03 \pm 0.03$ |
| housing | $\mathbf{0.75} \pm 0.03$ | $0.65 \pm 0.07$ | $0.70 \pm 0.03$ | $0.71 \pm 0.03$ |
| iris | $\mathbf{0.81} \pm 0.09$ | $0.78 \pm 0.21$ | $\mathbf{0.81} \pm 0.08$ | $0.70 \pm 0.08$ |
| pendigits | $0.95 \pm 0.00$ | $0.82 \pm 0.06$ | $0.94 \pm 0.00$ | $\mathbf{0.96} \pm 0.00$ |
| segment | $\mathbf{0.95} \pm 0.01$ | $0.78 \pm 0.06$ | $0.94 \pm 0.00$ | $0.93 \pm 0.00$ |
| spo | $0.16 \pm 0.02$ | $0.07 \pm 0.03$ | $\mathbf{0.18} \pm 0.02$ | $\mathbf{0.18} \pm 0.02$ |
| stock | $\mathbf{0.77} \pm 0.05$ | $0.56 \pm 0.23$ | $0.75 \pm 0.03$ | $0.63 \pm 0.07$ |
| vehicle | $\mathbf{0.87} \pm 0.03$ | $0.69 \pm 0.09$ | $0.84 \pm 0.03$ | $0.79 \pm 0.03$ |
| vowel | $0.73 \pm 0.03$ | $0.70 \pm 0.03$ | $0.73 \pm 0.02$ | $\mathbf{0.74} \pm 0.02$ |
| wine | $0.94 \pm 0.03$ | $0.85 \pm 0.17$ | $\mathbf{0.96} \pm 0.03$ | $0.86 \pm 0.08$ |
| wisconsin | $0.75 \pm 0.03$ | $0.56 \pm 0.07$ | $\mathbf{0.78} \pm 0.03$ | $0.75 \pm 0.04$ |

Table 2: Spearman correlation on 21 datasets averaged by 10-fold cross-validation. The learning rate **and** the temperature $\varepsilon$ are chosen from $(10^{-3}, 10^{-2}, 10^{-1})$ by grid search over 5-fold cross-validation.

| Dataset | Perturbed FY | Perturbed + Squared loss | Squared loss | Gradient proxy [54] |
|---|---|---|---|---|
| authorship | $0.95 \pm 0.01$ | $0.93 \pm 0.02$ | $\mathbf{0.96} \pm 0.01$ | $0.76 \pm 0.04$ |
| bodyfat | $0.35 \pm 0.07$ | $0.34 \pm 0.08$ | $\mathbf{0.36} \pm 0.08$ | $0.32 \pm 0.07$ |
| calhousing | $\mathbf{0.26} \pm 0.02$ | $0.25 \pm 0.04$ | $\mathbf{0.26} \pm 0.02$ | $0.17 \pm 0.06$ |
| cold | $0.08 \pm 0.04$ | $0.08 \pm 0.04$ | $\mathbf{0.09} \pm 0.04$ | $0.04 \pm 0.04$ |
| cpu-small | $0.53 \pm 0.01$ | $\mathbf{0.54} \pm 0.02$ | $0.50 \pm 0.02$ | $0.50 \pm 0.01$ |
| diau | $\mathbf{0.26} \pm 0.03$ | $\mathbf{0.26} \pm 0.02$ | $\mathbf{0.26} \pm 0.03$ | $0.25 \pm 0.03$ |
| dtt | $0.14 \pm 0.04$ | $0.13 \pm 0.04$ | $\mathbf{0.15} \pm 0.04$ | $0.11 \pm 0.04$ |
| elevators | $\mathbf{0.80} \pm 0.01$ | $0.79 \pm 0.01$ | $0.77 \pm 0.01$ | $0.76 \pm 0.02$ |
| fried | $\mathbf{1.00} \pm 0.00$ | $0.99 \pm 0.01$ | $0.99 \pm 0.00$ | $\mathbf{1.00} \pm 0.00$ |
| glass | $\mathbf{0.88} \pm 0.05$ | $0.84 \pm 0.06$ | $0.86 \pm 0.06$ | $0.83 \pm 0.05$ |
| heat | $\mathbf{0.06} \pm 0.03$ | $0.05 \pm 0.03$ | $\mathbf{0.06} \pm 0.02$ | $0.03 \pm 0.03$ |
| housing | $\mathbf{0.76} \pm 0.03$ | $0.75 \pm 0.03$ | $0.70 \pm 0.04$ | $0.71 \pm 0.03$ |
| iris | $0.80 \pm 0.12$ | $\mathbf{0.86} \pm 0.11$ | $0.81 \pm 0.08$ | $0.70 \pm 0.08$ |
| pendigits | $\mathbf{0.96} \pm 0.00$ | $0.95 \pm 0.00$ | $0.94 \pm 0.00$ | $\mathbf{0.96} \pm 0.00$ |
| segment | $\mathbf{0.95} \pm 0.00$ | $0.94 \pm 0.01$ | $0.94 \pm 0.01$ | $0.93 \pm 0.00$ |
| spo | $\mathbf{0.18} \pm 0.02$ | $\mathbf{0.18} \pm 0.02$ | $\mathbf{0.18} \pm 0.02$ | $\mathbf{0.18} \pm 0.02$ |
| stock | $\mathbf{0.78} \pm 0.07$ | $0.70 \pm 0.21$ | $0.75 \pm 0.03$ | $0.63 \pm 0.07$ |
| vehicle | $\mathbf{0.89} \pm 0.02$ | $0.86 \pm 0.03$ | $0.84 \pm 0.03$ | $0.79 \pm 0.03$ |
| vowel | $0.74 \pm 0.02$ | $\mathbf{0.75} \pm 0.03$ | $0.73 \pm 0.02$ | $0.74 \pm 0.02$ |
| wine | $0.95 \pm 0.04$ | $0.91 \pm 0.07$ | $\mathbf{0.96} \pm 0.04$ | $0.86 \pm 0.08$ |
| wisconsin | $\mathbf{0.78} \pm 0.03$ | $0.77 \pm 0.03$ | $0.77 \pm 0.03$ | $0.75 \pm 0.04$ |

Figure 7: As in Figure 4, we show here both (**Top**) the average number of instances with exactly correct ranks (perfect ranks) for all 100 labels (**Bottom**) the average number of correctly ranked labels (partial ranks). In both for different values of $\sigma \in [10^{-2}, 10^3]$, for four methods.

Figure 8: We report the same metrics as in Figure 7 of perfect ranks and partial ranks at two fixed noise levels, as a function of the number of epochs **Left.** for $\sigma \approx 0.1334$ **Right.** for $\sigma \approx 0.7449$.