[Reviews · NeurIPS 2020]

Review 1

Summary and Contributions: This paper employs perturbations of max/argmax functions to enable differentiation over the piece-wise constant functions. This enables backpropagation for common operations like sorting/ranking and min cost path planning.

Strengths: The approach appears to be quite general and can be applied to many discrete optimization problems without substantial difficulty. The experimental results demonstrate significant benefits on three different applications, showing the general utility of the approach.

Weaknesses: The main described difference from SparseMap is the ability to trivially parallelize the proposed approach while SparseMap is sequential. While important, this alone does not seem to disqualify SparseMap as an appropriate baseline for comparison.

Correctness: It is unclear where O(|E|) time complexity for shortest path (with real-valued costs) comes from. Are there additional assumptions?

Clarity: The paper is clearly written.

Relation to Prior Work: The benefits compared to more general frameworks (e.g., based on differentiation through a QP or conic/convex optimizations) and SparseMAP should be further explained. Some of these are discounted as problem-specific; what are the implications for label ranking and shortest path?

Reproducibility: Yes

Additional Feedback: POST-RESPONSE: Thank you for addressing my concerns.


Review 2

Summary and Contributions: The authors contribute to the theory of embedding discrete optimizers (combinatorial solvers) into neural networks. The main technical challenge there is "differentiation" of piecewise constant mappings induced by such solvers. The authors propose (a) a scheme that provides an unbiased gradient estimate of "smoothening" (in the sense of convolution) of the piecewise constant function and (b) a Young-Fenchel loss function to employ. They provide extensive evaluattion.

Strengths: The topic is of very high relevance and I consider both contributions as significant to the community. The connections with existing theory are appealing. Experimental performance looks promising.

Weaknesses: (a) there are some omissions and inaccuracies in the experimental evaluation which make it hard to assess its quality (see below) (b) More importantly, there is a somewhat serious misconception throughout the presentation of the paper. There are two different tasks when it comes to embedding solvers to neural networks: Task 1 - Gradient estimation (GE): This precisely captures how the piecewise constant aspect is handled. Methods that estimate gradient then allow complete flexibility of the layers that precede the solver and the layers that follow (e.g. the loss function. Task 2 - Loss Functions (LF): In the specific and relatively common setup, when the solver is *the last layer* AND the ground truth solver outputs *are known*, it is desirable to design good loss functions that give strong gradient signal and have good impact on generalization. These two tasks are related but they are fundamentally different. First of all, the (LF) task has solutions going back to 2005, such as the max-margin loss [L1, L2], which is being used even in very recent architectures [L3]. The underlying theme is to introduce some sort of robustness. Either by margin such as the max-margin loss, or by noisifying inputs such as the suggested YF loss. On the other hand, Vlastelica et al is a pure (EG) method (no loss function is suggested, only naive Hamming loss is used). It also has an application in which the solver output undergoes some postprocessing [L4] -- something unreachable for (LF) methods. Another way to ilustrate the difference is that the naive loss used in Vlastelica et al, was in further applications [L4, L5] appended with a margin for increased performance, precisely in the spirit of (LF) methods. Now back to this work. Both the theory and the experimental parts read as if the two tasks were conflated. The claimed contributions revolve mostly around (GE) but almost all experiments use the (LF) contribution - and do not touch Proposition 3.1. (the key for GE). I do not understand this. [L1] Tsochantaridis et al. Large Margin Methods for Structured and Interdependent Output Variables [L2] Franc et al. Discriminative Learning of Max-Sum Classifiers [L3] Sadat et al. Jointly Learnable Behavior and Trajectory Planning for Self-Driving Vehicles [L4] Rolinek et al. Optimizing Rank-based Metrics with Blackbox Differentiation [L5] Rolinek et al. Deep Graph Matching via Blackbox Differentiation of Combinatorial Solvers

Correctness: I found the mathematical claims correct. The empirical methodology is however again very confusing and at times just wrong. The authors very consistently write about "blackbox loss of Vlastelica et al" but *there is no such thing*. Again, Vlastelica et al, is a pure (GE) method and does not propose any loss function. In general comparing the YF-loss to Vlastelica et al. is a bit of apples to oranges (LF) to (GE). Maybe a fairer comparison would be Vlastelica et al + margin [L4, L5] where YF could additionally claim more simplicity and one fewer hyperparameter. What is certainly missing is a comparison of YF-loss to the classical max-margin loss (this is apples to apples). And again, from carefully inspecting the experimental results - including the supplementary - I believe that both proposed methods empirically show promising performance and would also perform equal/better in the correct comparisons. My beef is purely with the experimental methodology. Additional remarks: The curves reported in Figure 6 are *wildly* inconsistent with numbers reported in Vlastelica et al. I will assume good intent and take a guess: 1) The labels for bb-loss and squared loss are switched 2) The metric reported is (slightly) different to the one Vlastelica et al. I did not manage to find the value of M for the ranking experiment. This is particularly unfortunate since it is the only experiment that uses Proposition 3.1 to compute the Jacobian. I would love to know if the high variance Jacobian estimate for M=1 can lead to stable training. For the combinatorial applications and the whole (GE) branch of this paper, I find it critical.

Clarity: The technical parts are written clearly and are easy to follow. I have some issued with the overall presentation as written above.

Relation to Prior Work: Discussion of the max-margin loss [L1, L2] should certainly be added. There is some missing literature regarding learning to rank. For one, it would be fair to mention that Vlastelica et al. has a follow-up work focusing on ranking [L4]. Also, there are at least [L6, L7] and others. Obviously, there are also many works aimed to "differentiable top-k" [L8, L9] for start or even surveys of classical methods [L10]. Acknowledging some would be appropriate Given that most of the experimental section revolves around comparing to Vlastelica et al, it is strange that in the introduction it is presented with one sentence that revolves around a particular technical aspect rather than the point of the method. [L6] Cuturi, et al. Differentiable Ranks and Sorting using Optimal Transport [L7] Blondel, et al Fast Differentiable Sorting and Ranking [L8] Xie et al. Differentiable Top-k Operator with Optimal Transport [L9] Amos et al. The Limited Multi-Label Projection Layer [L10] Lapin et al, Loss Functions for Top-k Error: Analysis and Insights

Reproducibility: Yes

Additional Feedback: I think the raw technical contribution of this work is 100% publishable at a top tier conference. However, I cannot back the current version of the writeup for the reasons sketched above. ========== POST REBUTTAL =========== The authors addressed many of my issues and even though, I would be more comfortable approving the updated manuscript, I am ready to take a leap of faith :). Overall, this work provides alternatives to existing methods for incorporating discrete optimizers to NNs. This is worth acceptance. During the first read, I had the impression that the paper tries to position itself as *finally allowing solvers inside networks* which is not accurate. I believe this will be toned down in the final version. One other ongoing issue is that the (GE) part of the paper was never tested in the case M=1. I find this essential for incorporating possibly expensive discrete optimizers. The competing (GE) method of Vlastelica et al by default operates with M=1. I updated my score.


Review 3

Summary and Contributions: The paper proposes a simple yet deeply insightful method for end-to-end learning with discrete optimization layers (argmax of an LP) via stochastically perturbed optimizers. This can be thought of as extending the recently line of differentiable (smooth) optimimization work to cases where a useful true gradient does not exist (i.e., cases where the gradient of the solution map is zero almost everywhere). It can also be seen as a generalization of softmax for discrete classification problems. The work builds on well-established literature and demonstrates the application of the proposed method on some toy experiments.

Strengths: + Simple method that allows discrete optimization layers to be incorporated into end-to-end learnable models + The method does not require knowledge of the solve (i.e., efficient blackbox solvers can be used unmodified) + The paper is very clearly written and contains insightful remarks and succinct examples.

Weaknesses: I'm nitpicking here since the paper is actually very strong: - The experiments are run on toy examples (which is okay since the main contributions are theoretical). While they validate that the approach can be applied to problems it's unclear that the method outperforms other (ad hoc) approximation schemes on larger problems. - The running time is also not reported. Is it the case that running time scales linearly with M? Section 3 discusses warm-starts but it's not clear that this actually helps for discrete optmization problems where the solution may make a large jump from the previous solution. - Permutation learning has been explored using the Sinkhorn algorithm in the machine learning literature. It would be interesting to compare the proposed method (and results from Section 5.2) to those existing methods.

Correctness: The claims appear to be correct. Proofs are included in the supplementary material.

Clarity: The paper is very well written.

Relation to Prior Work: Links to related work are clear.

Reproducibility: Yes

Additional Feedback: The phrase "the computational graph is broken" on L186 is not quite right. The graph is perfectly fine for the forward pass (i.e., inference). The only problem is that the exact gradient is not useful for learning. Perhaps it would be better to say that the graph provides no information during the backward pass for learning. POST REBUTTAL: The rebuttal addresses my concerns and appears to address the concerns of other reviewers. Perhaps some of the prior work raised by other reviewers can be incorporated into the final revision. I maintain my initial recommendation.


Review 4

Summary and Contributions: The submission describes a framework for differentiating through optimization by solving perturbed optimization problems and averaging the results. Specifically, the paper considers differentiating through convex optimization problems with linear objectives. The optimal objective is smoothed by randomly perturbing the linear objective, solving the problem for each perturbed objective, and taking the expectation over the perturbations. It is then shown that the derivative of the smoothed problem with respect to the linear objective, as well as the derivative of the argmax of the smoothed problem, can be evaluated as expectations of functions of the perturbed solutions. A novel kind of structured loss is also derived leveraging these results. The gradient of the loss is easily computed, although the exact form of the loss is not explicit. Connections to Gibbs distributions and the Gumbel-max trick are also discussed. The method can be interpreted as generalizing the Gumbel-max trick.

Strengths: 1— provides a simple and clean framework for differentiating through optimization Although various ways of differentiating through optimization problems have been proposed previously, the one proposed in the submission is attractive because it is conceptually simple, simple to implement, and covers a wide variety of problems with a clean framework. This passes the test of being something that I might conceivably want to try at some point in the future, which increases the likelihood that it will have a significant impact. 2— the Fenchel-Young loss and connections to Gibbs distributions are interesting I found the connections to Gibbs distributions and the Gumbel-max trick to be particularly interesting. The parameter gradient of the Gibbs distribution has a well-known form as a difference between the expected features under the model distribution and the observed features. The Fenchel-Young loss seems to generalize this concept, in the sense that the expected features under the model are replaced with an expectation of perturbed argmaxes. When the perturbations are Gumbel, then this reduces to the Gibbs case. This raises the possibility of thinking about ways in which algorithms for learning in Gibbs distributions, which tend to suffer from intractable inference, could be relaxed a bit by reformulating them as perturbed optimizations with different types of noise. This is a bit speculative, since the paper ostensibly only covers cases where inference reduces to convex optimization (i.e., where inference is already tractable), but the idea serves to show how this work might spark follow-on research. 3— the method is rigorously analyzed The paper is careful about stating the differentiability and uniqueness properties of the method, and additionally provides a convergence rate for the consistency of estimating parameters based on the Fenchel-Young loss.

Weaknesses: 1— the main point is a bit unclear See discussion in “clarity” section. 2— it’s hard to separate what’s novel from what’s not novel The middle of the paper is filled with a barrage of facts, many of which are results from other papers. This makes it hard to figure out exactly what part of the work is novel. Perhaps the contribution is mainly just synthesizing these facts into a clean framework to solve a particular class of problems. If so, that’s ok, but that should really be made clearer. 3— advantages compared to comparable methods are unclear The submission has one potentially significant technical deficiency, which is that it’s unclear exactly what the advantages of this method are compared to competing methods. In particular, I think both finite-differencing and work on differentiating through the argmax of convex programs are relevant. Finite-differencing can also be thought of as a perturbation method, except one where the perturbations are chosen deterministically. The resulting gradient is biased, due to the finite step size, but there is no variance in the estimate, since the perturbations are deterministic. This method, on the other hand, has both bias (from smoothing the problem) as well as variance from the stochasticity of the perturbations. One could argue that the smoothing is the important part, but if that’s so, then why not apply different kinds of smoothing before finite-differencing? Overall, it’s not obvious to me why this method would actually be superior in practice to finite-differencing. Some discussion and experiments would be reassuring. Prior work on differentiating though the argmax of convex programs is at least mentioned, but it’s still unclear to me why this proposal would generally be superior. One can smooth a general convex optimization problem by penalizing the constraints with a log-barrier of adjustable strength (as in ref [A], for example). Adjusting the barrier strength may address the cited issue of zero Jacobians. In any event, it would be useful to see comparisons to at least some comparable baseline. The experiments are very weak in this regard. [A] Schulter, Samuel, et al. "Deep network flow for multi-object tracking." Proceedings of the IEEE Conference on Computer Vision and Pattern Recognition. 2017.

Correctness: I detected no issues with the correctness of the claims.

Clarity: Perhaps the greatest weakness of the submission is that the main point is a bit unclear. The body of the paper gets lost in numerous, miscellaneous, random technical properties of the method, and I as a reader was often left wondering what I should be taking away from the paper. If I were to rewrite the paper, I would focus more on answering the question, “why should I use this method compared to comparable methods?” Also, “what insights does this new viewpoint provide, /and/ why do those insights matter?” Although there is some discussion of insights, the second half—i.e., the importance of those insights—is lacking. For example, the whole discussion about the convex regularizer and its role in the Fenchel-Young loss is very confusing, because it is never acknowledged that this function generally has no interpretable form, since it is expressed only as the Fenchel conjugate of some complicated function. So, while it’s nice to know that you I minimize the loss, I don’t have any intuition for what loss I’m minimizing, besides “it’s the Bregman divergence of some black-box function.”

Relation to Prior Work: As mentioned elsewhere in this review, I thought that methods based on differentiating through convex programs were perhaps unfairly dismissed. I think the paper would benefit from an experimental comparison to at least one such method.

Reproducibility: Yes

Additional Feedback: POST-REBUTTAL COMMENTS The rebuttal was helpful to clarify some issues. My main take-away is that there are really two problems: smoothing and the differentiation method. e.g., one could smooth with randomization, and then differentiate with stochastic finite differences; or smooth with an analytic method (e.g., log-barrier) and then differentiate analytically. It's still a little unclear to my why this method would be better than the former approach. Overall, though, I still think this is a general, clean and simple way to differentiate through optimizers. My main concern about the paper is still the presentation, which could use a lot of work. Maybe thinking about the method in terms of those two axes (smoothing method and differentiation method) might help clarify the presentation a bit. Again, I think it would be helpful to ask questions like, "what properties of the method are relevant to its practical applicability?" and organize the paper in those terms, instead of presenting a random barrage of facts with little context. At the very least, I hope pages 4 and 5 will be revised along these lines--again, they should be organized around high-level questions that are of practical significance to provide context, with the technical properties providing answers to these questions. For example, a toy example could be used to motivate the problem of the Jacobian being zero almost everywhere. How does this practically appear in competing methods? Then, how can we be reassured that this is not a problem in this method, both through a tangible example (e.g., a figure) and through analysis? I think revising the paper along those lines would raise the profile of this work considerably.

[Author Response · NeurIPS 2020]

We would like to thank the reviewers for their overall positive assessment of our work, and for their detailed comments.

**Reviewer 1:** SparseMap as an appropriate baseline We agree. We will add it to sec. 5.1 and 5.2, similar to Figure R1 below. A potential drawback compared to our method is the possibility of zero gradients. $O(|E|)$ time complexity for shortest path You are correct, this requires integer costs. For general costs, we will replace it with $O(|V|^2)$, for Dijkstra's algorithm. benefits compared to more general frameworks [...] further explained. We will detail further the benefits: guarantee of non-zero gradients and ease of implementation (simple averages of solutions from the same solver).

**Reviewer 2:** On the tasks of gradient estimation (GE) and loss functions (LF), and additional references suggested Thank you for the additional references, we will include them with discussion. We tried our best to distinguish GE from LF throughout the paper but we will further clarify, following this perspective. We consider this work to provide advances for both of these tasks: Our perturbed optimizers can be used as intermediate layers, with any loss, and are well-defined without requiring a loss (GE); they also naturally yield a FY loss (LF) at the last layer, for which gradient estimation is even easier. We focused mostly in our experiments on cases showcasing both our contributions, with perturbed optimizers being used in a FY loss, but not exclusively, as you point out. We will also include (Fig R1) (in addition to Figure 3), showing that on the 21 dataset experiment, for a given loss ($\ell_2$), our GE method with perturbation is on par or better than the GE method of Vlastelica et al. in terms of performance, while there is consistent performance between $\ell_2$ and FY losses using our GE method.

*Figure R1: zoom for details*

about "blackbox loss of Vlastelica. et al" but *there is no such thing* We agree, and will change the designation to "Gradient estimation for loss (eqref)." comparing the YF-loss to Vlastelica. et al. is a bit of apples to oranges (LF) to (GE). We argue that it is fair to compare GE and LF if the end task is the same, especially when replicating an experiment. However, we agree that comparing different GE methods that are generic is interesting and will add Figure R1 in this respect. comparison of YF-loss to the classical max-margin loss We will also include it in experiments The curves reported in Figure 6 are *wildly* inconsistent [...] 1) The labels for bb-loss and squared loss are switched The labels were indeed accidentally swapped between bb-loss and squared loss in the left part of Figure 6, thanks for catching it. We will correct this. 2) The metric reported is (slightly) different to the one Vlastelica et al. Following this paper, we reported the proportion of paths with exactly "the optimal costs" with all methods. In order to display the evolution of this metric (and cost ratio to optimal) along epochs, we ran an experiment using the code provided by the authors, with the same parameters. We report directly those figures. We found that another statistic "below 1% accuracy" was close to the numbers reported in this paper, but this is not one of the metric that we considered in our experiment. value of M for the ranking experiment $M = 10$ in the experiment on perturbed GE for the $\ell_2$ loss. Prior work being added Thank you for these references, we will cite them and include some discussion.

**Reviewer 3:** running time scales linearly with M? We will add running times and specify the hardware setup. Regarding the dependency in $M$, when running these on GPU, by the parallelizable nature of our technique, the running time is almost independent of $M$ (up to $M = 1000$). Further, in many cases (as discussed l.210-212, 250-251), the value of $M$ used in learning tasks to minimize the FY loss is very small (of order 1 to 10). Permutation learning with the Sinkhorn algorithm We will include a comparison with this benchmark for the ranking experiments, as in Figure R1. The phrase "the computational graph is broken" on L186 is not quite right. Thank you for pointing it out, we will add this remark.

**Reviewer 4:** Comments about clarity of the paper Regarding contributions, our propositions are novel, except Prop.3.1, as referenced, and Prop.2.1 (a few lines from classical results). If the paper is accepted, we will use the ninth page to describe section 2 and 3 in more ample details. We made efforts in the abstract and introduction to explain the application to learning tasks, and will use this space to detail it further, early in the paper. Comparison to differentiation of convex programs and finite differences We would like to clarify that we mainly focus on problems where the Jacobian is almost everywhere null. Hence finite-differencing or implicit differentiation of convex functions might not be useful if the function is not smoothed. Our approach provides automatically a smoothing of the function and an unbiased stochastic estimate of the corresponding Jacobian - very easy to implement and fast to compute. It is also very generic and does not require problem-specific adaptation. One can smooth a general convex optimization problem by penalizing the constraints with a log-barrier Thank you for the reference, we will include it. This could raise some algorithmic issues, requiring ad hoc solvers, instead of the existing ones. It is also not always tractable, for problems with efficient solvers but exponentially many facets in the polytope, a barrier approach is impossible. Our method does not suffer from this, and only requires to use the original solver, without a new, often slower method. A notable exception is optimal transport, which we mention, for which the entropic regularization yields an algorithmic advantage compared to the LP. Role of the convex regularizer We will make this point clearer: this formulation leads to many of the important properties of the perturbed optimizer (see Prop 2.2, 2.3, 4.1), including that $y_\varepsilon^*$ is one-to-one, implying that estimating $\theta$ from the observed model is possible. We will also further address the interesting phenomenon that you describe: the FY loss need not to be computed in order to be minimized. We perceive this as an advantage, since the gradients are very simple to compute, and it performs very well on several learning tasks (see Section 5), but deserves more explanation.

[Meta-Review · NeurIPS 2020]

The reviewers felt the paper describes valuable contributions, but were concerned that the presentation had issues such as clarity. The author response helped explain some technical ideas, and the reviewers agreed that the authors should be able to make a minor revision to make a solid conference paper.